# Lipid Profile of *Larix cajanderi* Mayr in Adaptation to Natural Conditions in the Cryolithozone

**DOI:** 10.3390/ijms26010164

**Published:** 2024-12-28

**Authors:** Vasiliy V. Nokhsorov, Tatiana D. Tatarinova, Lyubov V. Dudareva, Natalia V. Semenova, Trofim C. Maximov

**Affiliations:** 1Institute for Biological Problems of Cryolithozone, Siberian Branch of Russian Academy of Sciences, Division of Federal Research Centre “The Yakut Scientific Centre of the Siberian Branch of the Russian Academy of Sciences”, 41 Lenina Av., Yakutsk 677000, Russia; t.tatarinova@gmail.com (T.D.T.); tcmax@mail.ru (T.C.M.); 2Siberian Institute of Plant Physiology and Biochemistry, Siberian Branch of Russian Academy of Sciences, 132 Lermontova Str., Irkutsk 664033, Russia; laser@sifibr.irk.ru (L.V.D.); tashasemyonova@mail.ru (N.V.S.)

**Keywords:** adaptation, cryolithozone, fatty acids, GC–MS, lipids, *Larix*, membrane lipids, polar lipids, shoots, ∆5-UPIFA

## Abstract

The prevalence of coniferous trees in the forest landscapes of northeastern Siberia is conditioned by their high frost resistance. The Kajander larch (*Larix cajanderi* Mayr), which can survive under natural conditions (down to −60 °C) in the cryolithozone of Yakutia, is the dominant forest-forming species. We hypothesise that our study using HPTLC–UV/Vis/FLD, TLC–GC/FID, and GC–MS methods of seasonal features of the lipid profile of Kajander larch tissues will bring us closer to understanding the mechanisms of participation of lipid components in the adaptation of this valuable tree species to the cold climate of the cryolithozone. Rare delta5-unsaturated polymethylene-interrupted fatty acids (∆5-UPIFA) were identified in the fatty acids (FAs) of *L. cajanderi* shoots, including 18:2(Δ5.9) (taxoleic), 18:3(Δ5.9.12) (pinolenic), and 18:4(Δ5.9.12.15) (coniferonic). It was found that the content of ∆5-UPIFA in *L. cajanderi* shoots markedly increased (1.5-fold, representing up to 23.9% of sum FAs) during the autumnal transition of trees to dormancy. It was observed that the ranges of low temperatures experienced during the prolonged winter period primarily determined the structural diversity of membrane lipids and their constituent FAs during the cold adaptation of *L. cajanderi*. The results obtained can be used for the selection of molecular markers of cold tolerance in woody plants, including fruit trees.

## 1. Introduction

The forest-forming plants that grow in the cold climate and permafrost of Central Yakutia are distinguished by a remarkable frost resistance. The viability and species composition of the dendroflora in the cryolithozone are determined by extremely low air temperatures (down to −60 °C) and the absence of recurrent warming pulses throughout winter. The ability of woody plants to resist unfavorable environmental factors, including low temperatures, is the result of physiological and biochemical alterations, particularly structural and chemical rearrangements in cell membranes. In response to stressful environmental conditions, coniferous plants utilize a range of defensive mechanisms involving lipids and the associated fatty acids (FAs), both being integral components of cell membranes. The current understanding of plant adaptation to temperature change suggests that one of the primary adaptive strategies employed is a reduction in membrane lipid unsaturation at higher temperatures and an increase at lower temperatures [1].

The spatial structure of the lipid bilayer of cell membranes is largely determined by the composition and degree of desaturation of the fatty acids in the lipids. The adaptive potential of plants significantly depends on their capacity to maintain membrane fluidity and prevent lipid phase transition when exposed to stressors [2,3].

Abiotic stresses, such as drought and low temperature, induce a variety of lipid-dependent signaling responses in which lipids serve as mediators to mitigate stress responses in plant cells and activate defense systems. Consequently, signaling lipids, including phosphatidic acid (PA), phosphatidylinositol (PI), sphingolipids, lysophospholipids, oxylipins, and N-acylethanolamines, are induced in response to stressors of diverse natures [4,5,6,7]. The principal lipids of plasma membranes, mitochondrial membranes, and microsomes of plant cells are phosphatidylcholine (PC) and phosphatidylethanolamines (PEs). It has been demonstrated that adaptation to low temperatures is accompanied by an increase in lipids that destabilize the bilayer membrane, including PEs and PA [8], and an increase in unsaturated fatty acids, which lowers the phase transition temperature [9]. For example, the greatest increase in PC during cold adaptation was found in the needles of balsam fir [10], as well as in other conifers such as *Pinus sylvestis* L., *Picea obovata* Ledeb. [11,12], and *Pinus nigra* J.F. Arnold [13].

Membrane lipids are critically important under stress conditions to preserve the lamellar structure of the chloroplast, stabilize chloroplast membranes, facilitate the membrane protein packing, and control membrane fluidity by regulating fatty acid desaturation [14,15,16,17]. In this respect, less is known about the role of chloroplast membrane lipids, which are mainly composed of glycolipids (GLs), sulfolipids, and phosphatidylglycerol (PG). Sulfolipids, specifically sulfoquinovosyldiacylglycerol (SQDG), are predominantly localized in the lamellae of chloroplasts. Monogalactosyldiglyceride (MGDG) is localized in thylakoid membranes and is associated with photosystems I and II. Digalactosyldiglyceride (DGDG) is the major lipid component of both thylakoid membranes and the outer envelope membrane of chloroplasts. It has been demonstrated that DGDG is required in substantial quantities during the active growth and development of plants [18].

In permafrost ecosystems, the Yakutian population of *L. cajanderi* is the best adapted conifer to grow in the cryolithozone, being the most resistant to low negative temperatures. However, the biochemical mechanisms of low-temperature adaptation of *L. cajanderi* to the natural condition of Yakutia have not yet been studied. Data are now available on the peculiarities of the lipid composition of needles in tissues of embryogenic and non-embryogenic cell lines of *Larix sibirica* Ledeb., where β-sitosterol dominates in free and bound sterols [19]; the composition of phospholipids and neutral lipids has also been studied [20,21]. The FA profile of conifer seed lipids is known to contain unusual ∆5-UPIFAs with *cis* (Z)-configuration: 18:2(Δ5.9) (taxoleic), 18:2(Δ5.11) (ephedrenic), 18:3(Δ5.9.12) (pinolenic), 18:4(Δ5.9.12.15) (coniferonic), 20:1(Δ5.11) (keteleeronic), 20:3(Δ5.11.14) (sciadonic), and 20:4(Δ5.11.14.17) (juniperonic) [22,23]. Previously, it was shown that the content of delta5-unsaturated polymethylene-interrupted fatty acids (Δ5-UPIFA) in the pine needles of one of the representatives of Pinacea—pine native to Eastern Siberia—could reach up to 20.4% of the sum of FAs in the case of decreasing ambient temperature [24]. The abundance of these acids and α-linolenic acid (ALA, 18:3n−3) in chloroplast membrane lipids in pine and other conifer needles is typically ascribed to plant adaptation to low temperatures [24].

The specific role of lipid molecules, their metabolic mechanisms, and their involvement in the interaction between plant organisms and their environment remain incompletely understood. In this context, it seems highly relevant to investigate the adaptive modifications of lipids and their constituent FAs in the processes contributing to the low-temperature resistance of *L. cajanderi* in the cryolithozone.

In this context, knowledge and study of the mechanisms of cold adaptation in conifers under natural growing conditions is of both theoretical and practical interest. For example, as climate instability increases, there is a growing need to identify plant species and genotypes that are best adapted to increased frost and drought resistance, as well as resistance to pests and diseases. Consequently, changes in the lipid composition of tree membranes, including conifers, can be considered as molecular markers of cold tolerance, which are promising both for the selection of high-yielding and stress-adapted crops, such as fruit crops, in biotechnology, and for maintaining the stability of boreal ecosystems in the forestry practice. Aims of this study: to analyse the lipid profile in the tissues of Kajander larch in the annual cycle, and to identify features of lipid metabolism that contribute to the formation of such (unique) resistance of larch during its adaptation to specific cryolithozone conditions.

## 2. Results

### 2.1. Seasonal Changes in Air Temperature and Physiological Phases of L. cajanderi Development

The object of this study was the Kajander larch (*Larix cajanderi* Mayr), a boreal East Asian species, and the main forest-forming species of the cryolithozone of Yakutia and northeastern Siberia (Figure 1A). The trees are found in the forest park zone of the Botanical Garden of IBPK SB RAS (Institute for Biological Problems of Cryolithozone SB RAS). One-year-old shoots without needles (Figure 1B) were sampled once a month during the year.

Plants growing in the climate of Central Yakutia, which include ultra-low winter temperatures and the presence of relict permafrost at depth (cryolithozone), are subjected to the most significant stress factors. The sharp continental climate of Central Yakutia (Yakutsk) is distinguished by extreme conditions. The average monthly air temperature in January is −38.6 °C within the range from −35 °C to −42 °C (without recurrent winter warming pulses), while in July it varies from +13 °C to +26 °C (Figure 2). Total annual precipitation typically ranges from 230 to 270 mm (data from the Weather and Climate website http://www.pogodaiklimat.ru, accessed on 17 September 2024). Permafrost, or cryolithozone, occurs across the entire territory.

The maximum average monthly air temperature of 19.9 °C in the cryolithozone of Yakutia is observed in July. In September, the average monthly air temperature decreases to 6.4 °C, accompanied by a reduction in daylight hours, which induces a state of deep physiological dormancy in *L. cajanderi*. This process is accompanied by the fall of needles, since, unlike evergreen conifers, *L. cajanderi* sheds needles for winter dormancy, thereby increasing its cryotolerance by reducing water loss through winter transpiration. October is typified by stable negative air temperatures (–6.9 °C) and persistent snow cover. In November, the air temperature drops to −25.9 °C, which marks the onset of induced dormancy in conifers. In December and January, low air temperatures are observed in the cryolithozone of Yakutia, with the average monthly temperature reaching as low as −37 °C. The spring season is marked by significant fluctuations in daily temperature. The mean air temperature in March is recorded at −19.1 °C, while in April it drops to −3.7 °C. In May, the temperature rises to 8 °C. With higher temperatures and a longer photoperiod in late May, current-year needle formation starts in *L. cajanderi* shoots.

### 2.2. HPTLC Profile of Polar Lipids in L. cajanderi Shoots Within a One-Year Cycle

The seasonal dynamics of membrane lipid content in *L. cajanderi* shoots was investigated using HPTLC–UV/Vis/FLD (Figure 3). Quantification of polar lipid (PL) content was performed in daylight, following derivatization (by HPTLC-Vis) (Figure 3A). Visualization of lipids by HPTLC-FLD at 366 nm (Figure 3B) revealed the presence of other lipophilic compounds on the plate. However, when the plates were visualized by HPTLC–UV at 254 nm (Figure 3C), no PLs were evident. We found that throughout the annual cycle of *L. cajanderi* in all seasons, major lipids were represented by MGDG, DGDG, SQDG, PE, and glycoceramides (GlCer1 and GlCer2), whose content exceeded 10% of total lipids. At the same time, PC, PA + PG + diphosphatidylglycerol (DPG), phosphatidylserine (PS), and PI, referred to as minor lipids, accounted for less than 10% of total lipids in *L. cajanderi* shoots (Figure 4). Of greatest interest were membrane lipids with a significant seasonal variation in content, as well as lipids whose accumulation was observed during preparation of plants for dormancy, which is likely related to the formation of low-temperature resistance in these plants.

The content of neutral lipids (NLs) in the autumn–winter months significantly decreased, up to 2.4 times, compared to the summer period. The results of these studies showed that spring shoots of *L. cajanderi* accumulated up to 20.5% of the sum of GlCer2. A sharp decrease in GlCer2 was revealed with the onset of summer. The lowest values (4% of sum) were observed in the autumn–winter period. The variation in dynamics of GlCer1 differed from that observed in GlCer2. Thus, GlCer1 increased in summer, when dry and hot weather persisted in the cryolithozone. The content of chloroplast lipid MGDG was low in the summer months, comprising no more than 10.1% of sum, compared to the levels observed in the autumn months (September–October, 18.4% of sum). In October and November, when growth processes were complete and the intensity of plant metabolism declined, the proportion of MGDG increased to 27.6%. At low negative temperatures in November, the level of another chloroplast lipid, DGDG, increased in the shoots of *L. cajanderi* up to 13% of sum. This then decreased in January by 5.6 times. High content of SQDG was observed during all of the seasons of this study. It was observed that from the end of spring (May) onwards, there was a gradual increase in SQDG until September–October when the plants underwent the stages of hardening to low temperatures. June shoots of the studied *L. cajanderi* accumulated PE up to 13.4% of the total. In October, with the onset of the autumn cold, the PE pool decreased by 3.4 times compared to the summer period, and such a low PE level was maintained until the end of the unfavorable conditions. A further increase in PE was observed to correlate with the rise in air temperature and light intensity in spring (April–May). Additionally, in *L. cajanderi* shoots exposed to stable low negative temperatures, there was a notable reduction in PA + PG + DPG (by 9.4 times) in comparison to summer. Conversely, spring air temperatures and a longer photoperiod resulted in a significant elevation in the PA + PG + DPG level. In September, there was a significant increase in PC (by 4.1 times) compared to August. Then, under negative air temperatures, the content of this lipid decreased to 0.6% of sum. In addition to the aforementioned alterations in the composition of membrane lipids, minimal PS and PI contents were consistently observed in *L. cajanderi* shoots under study during the annual cycle.

Since the PC/PE ratio has important physiological significance, we focused on these parameters in the lipid composition of *L. cajanderi* shoots. The PC/PE ratio decreased from January to the onset of spring, while a statistically significant increase of PC/PE was recorded in the autumn–winter period (Figure 5a). Autumn–winter periods were distinguished by a notable prevalence of bilayer lipids relative to non-bilayer PC + PI/PE + PA + DPG + DPG + PG in shoots of the trees under investigation (Figure 5b). Summer shoots of *L. cajanderi* exhibited higher DGDG/MGDG ratios than those observed during other periods (Figure 5c). Similar values were found for the ratio of all bilayer chloroplast lipids to MGDG (DGDG + SQDG + PG + DPG + DPG + PA)/MGDG ratio (Figure 5d).

### 2.3. Dynamics of FA Alterations in Membrane Lipids in L. cajanderi Shoots

The FA profile of membrane lipids (polar lipids) in *L. cajanderi* shoots revealed the presence of seven SFAs (14:0 to 23:0) and seven UFAs (18:1n−7 to 20:3∆5.11.14) (Table 1). The composition of membrane lipids also included the rarer ∆5-UPIFAs (18:2∆5.9; 18:3∆5.9.12; 20:3∆5.11.14), which collectively represented less than 9.2% of the total FAs. In the composition of PLs, linoleic acid (LA, 18:2n−6) and ALA were the most prevalent. The concentration of 16:0 FAs increased from spring to summer, whereas the concentration of LA and ALA showed a decline during this interval. The level of LA in the PLs of *L. cajanderi* shoots demonstrated an increase during the autumn–winter period. The ALA content exhibited a notable (1.2-fold) increase at the onset of autumn, in comparison to summer. This elevated level was sustained throughout winter and spring. In general, it was revealed that a 1.1-fold reduction in SFA in *L. cajanderi* shoots during the transition from summer to autumn led to an increase in the proportion of UFAs. The absolute content of FAs increased during the transition from summer to autumn, and significantly decreased in winter months at low temperatures compared to the autumn period. Additionally, the highest absolute total content of FAs in the composition of membrane lipids in *L. cajanderi* shoots was observed in spring, in comparison to other periods. The highest content of ALA (40.1% of the sum of FAs) was recorded in autumn.

### 2.4. Annual Dynamics in FAs in Total Lipids in L. cajanderi Shoots

The total lipid composition of *L. cajanderi* shoots revealed the presence of twenty-one fatty acids (Appendix A and Figure 6) with varying numbers of double bonds, including mono-, di-, tri-, and tetraene FAs (Figure 7). LA (up to 15.2 mg/g DW), which increased by 1.2 times from summer to autumn, dominated the total lipid composition. Among UFAs, a high content of 18:3(Δ5.9.12) was detected in L., reaching up to 9.1 mg/g DW in March shoots. On hot summer days, the amount of this acid decreased to 1.8 mg/g DW, then increased in response to a 3.6-fold decrease in ambient temperature in autumn, compared to the summer period values. In the shoots of *L. cajanderi*, the dominant saturated fatty acids were 16:0 and 22:0. The ∆5-UPIFA identified in total lipids were 18:2(Δ5.9) (taxoleic), 18:3(Δ5.9.12) (pinolenic), 18:4(Δ5.9.12.15) (coniferonic), and 20:3(Δ5.11.14) (sciadonic). The content of ∆5-UPIFA in *L. cajanderi* shoots increased by a factor of 1.5 (up to 23.9% of sum FAs) during the transition of the plants from summer to autumn. This level of ∆5-UPIFA was sustained until November. A slight decline to 19.0% of the total was shown in December and January, when low temperatures were recorded in the cryolithozone. The content of these compounds was found to increase once more in spring and then to decline gradually, almost 2-fold (12.8% of sum), with the onset of summer and the subsequent rise in air temperature. Additionally, very-long-chain fatty acids (VLCFAs), specifically 22:0 and 23:0, were identified in the total lipid composition in *L. cajanderi* shoots. The highest concentration of VLCFAs was observed during winter, with a mean value of 6.41 mg/g DW. With the onset of summer, the content of these components decreased by a factor of 2.1 in comparison to the winter season. In general, the total proportion of UFAs was found to exceed that of SFAs in the composition of total lipids in *L. cajanderi* shoots. A notable rise in the number of UFAs was noted in the early autumn months, coinciding with the onset of low quenching air temperatures and a reduction in daylight hours, marking the transition of the trees into physiological dormancy.

Additionally, we evaluated two integral indices of FAs unsaturation, specifically the degree of unsaturation and the number of double bonds (UI and DBI, respectively). These indices were assessed in *L. cajanderi* shoots across various seasons of the year (Figure 8). The results demonstrated that the degree of unsaturation of FAs in *L. cajanderi* shoots was slightly lower during winter compared to other seasons. DBI index values were found to be higher during the autumn months than in spring or winter. The synthesis of polyunsaturated FAs (PUFAs) is a series of sequential desaturation reactions of the FA acyl residue, with the end product of the previous reaction serving as a substrate for the subsequent one. The analysis showed that the activity level of ODR and LDR desaturase enzymes in *L. cajanderi* shoots remained almost unchanged during the annual cycle. The activity of Δ9-desaturases (SDR) decreased markedly towards summer compared to spring, but increased significantly in the autumn–winter period. During the transition from summer to autumn, the desaturation process markedly enhanced for 18-FA, along with a significant increase in the elongation of 16-FA.

The dataset of the seasonal dynamics of FAs in *L. cajanderi* shoots was subjected to principal component analysis (PCA) (Figure 9). Sample data were normalized using an internal standard (nonadecanoic acid, 19:0). Neither normalization nor transformation was applied to the data. Only automatic scaling was employed in the publicly available MetaboAnalyst resource (www.metaboanalyst.ca, accessed on 17 September 2024). The obtained FA profiles of *L. cajanderi* are shown to be clustered in September and October, which may indicate significant rearrangements of metabolism contributing to the preparation of the plants for dormancy with the onset of low quenching temperatures in September and persistent negative temperatures in late autumn. The score plot for June reflects an increase in saturated short-chain FAs in *L. cajanderi* shoots, including 14:0, 15:0, 18:0–i, and 16:1n−9. The deviations for components 1 and 2 were 29.8% and 12.2%, respectively.

## 3. Discussion

Cell membranes are a major site of low-temperature-induced damage in plants. Changes in the quantitative and qualitative composition of membrane lipids, specifically in relation to low-temperature tolerance in plants, play an important role in the molecular mechanisms underlying plant adaptation to cold. The formation of stable cell membrane structures during autumn and winter is a vital mechanism for woody plants exposed to extreme external conditions, as this ensures their stability and further vegetation. Membrane lipids and the associated FAs in overwintering organs (shoots) play an integral part in these processes, given that it is within these structures that embryonic organs and tissues are subjected to ultra-low negative temperatures (down to −60 °C and below) for extended periods [25]. The structure of all cell membranes is composed of GLs and phospholipids, which are amphiphilic lipids comprising a polar hydrophilic group and non-polar hydrophobic acyl chains. Sterols are also structural lipids that are present in the lipids of the membranes. Lipids function as a primary regulator of membrane fluidity, a crucial parameter for the optimal functioning of proteins, including those involved in transport systems. Typically, alterations in the lipid composition, degree of unsaturation, and length of the hydrocarbon chains of FAs are associated with the regulation of membrane fluidity [26]. It is the responsibility of UFAs to maintain a specific level of membrane fluidity at low temperatures, given that their phase transition temperature is markedly below that of physiological values [27]. The ratio of saturated to unsaturated fatty acids determines the phase transition point of biological membranes.

Membrane lipids are essential under stress conditions to preserve chloroplast lamellar structure, stabilize the chloroplast membrane [14], facilitate the packing of membrane proteins, and control membrane fluidity by regulating FA desaturation [15,16,17].

In addition to their status as components of cell membranes, PLs are known as important signaling molecules. In conjunction with other compounds, PLs regulate plant growth and development, as well as the cellular response to environmental changes [28]. It is typical for signaling lipids, including lysophospholipids, FAs, PA, diacylglycerol, oxylipin, sphingolipid, and N-acylethanolamine, to be present in plant tissues in low quantities [29]. The experiments demonstrated that seasonal alterations in the profiles of MGDG and DGDG were manifested in the tissues of *L. cajanderi* shoots throughout the annual cycle. These GLs maintain relatively high concentrations during winter, specifically from November to January, and until temperatures rise in spring. In contrast, we noted a significant reduction in MGDG and DGDG levels during the summer months. Similar results were obtained when studying the seasonal dynamics of GLs of balsam fir and pine needles [10,11]. The DGDG content increased during the winter months. It is thought that the increase in bilayer-forming DGDG during cold acclimation protects chloroplasts from winter damage by keeping the membrane more fluid [30]. DGDG is a chloroplast lipid that is essential for optimal plant growth [31]. On summer sunny days, active shoot growth and the formation of photosynthetic tissues, i.e., needles, typically take place. It is therefore probable that MGDG and DGDG are accumulated in the mesophylls of needles. Although GLs are not considered to be significant components of non-plastidial membranes, alterations in their concentration are frequently observed in response to stress conditions. This suggests that they play an important role in the development of stress tolerance in plants [32]. The PC/PE ratio is often employed to analyse membrane structures and functionality [33]. We found that this index in *L. cajanderi* shoots increased at low positive temperatures in September and decreased sharply at negative temperatures in October (Figure 5a). The highest PC/PE value is indicative of lamellar membrane structures, whereas a reduction in this ratio may suggest the formation of diverse hexagonal structures within the membrane, which could result in defects in membrane packing [34]. It is established that fluctuations in the DGDG/MGDG ratio influence the stability of the chloroplast membrane [35]. The DGDG/MGDG ratio is a key determinant of bilayer formation, as well as lipid–protein and interprotein interactions in thylakoid membranes. The present study revealed a notable shift in the DGDG/MGDG ratio, namely, an increase in MGDG (a lipid that does not form the bilayer) at the onset of persistent negative air temperatures in the cryolithozone in October and November. The quantity of MGDG in *L. cajanderi* shoots displayed an increase at autumn temperatures, which is likely attributable to the formation of chloroplasts and the development of their thylakoid system in the shoots. It can be posited that the maintenance of a high concentration of GLs in *L. cajanderi* tissues during October contributes to the accumulation of assimilates during the transition to dormancy, which in turn enables the overwintering and survival of woody plants in the cold of the cryolithozone of Yakutia.

It is acknowledged that exposure of plant cells to low air temperatures results in ultrastructural transformations associated with an increase in the size of chloroplasts and modification of mitochondrial shapes [36]. In addition to PC and PE, plant mitochondrial membranes contain PG, PI, and the mitochondria-specific lipid DGP [37,38]. In our experiments with *L. cajanderi* shoots, the increase in the content of PC (in September), PI, and the proportion of PG + DPG indirectly indicated significant changes in mitochondria induced by stressing factors in the autumn season and during the entrance into dormancy. During the onset of low temperatures in autumn under Nova Scotia conditions, the amount of phospholipids increased in balsam fir conifers, the predominant PL being PC, and an increase in PG and PI was also noted [10]. The amount of PI has previously been shown to increase in response to drought and salinity stress [39]. PI is the first signaling molecule in the phosphoinositide biosynthetic pathway [5]. During cold acclimation, PLs are often of interest due to shifts towards longer and more unsaturated FAs [40].

Comparative analysis of two larch species, *Larix gmelinii* var. japonica and *L. kaempferi*, revealed marked differences in the FA composition of the PL fraction (specifically phospho- and glycolipids) [41]. Thus, it was shown that in the needles, the ALA content in PLs reached up to 37.1% in *L. gmelinii var. japonica* and 45.1% in *L. kaempferi*. The data we obtained on the composition of FAs of the PL fraction in *L. cajanderi* shoots are consistent with the aforementioned results, as the ALA content in shoots was maintained at 40.8% of the FAs sum.

In spring, the content of the UFAs remained low, both in the composition of PL and total lipids in the overwintered shoots of *L. cajanderi*. This was evidently caused by high insolation, insufficient soil warming, and moisture deficit. Such conditions favor the accumulation of reactive oxygen species (ROS), which are generated during the resumption of photosynthesis [42]. These ROS may contribute to the photooxidation of unsaturated, especially polyunsaturated FAs of membrane lipids, and most likely unsaturated FAs that make up chloroplast lipids (MGDG, DGDG) [43]. Thus, the amount of DGDG and/or the DGDG/MGDG ratio are critical for supporting various cellular processes in the chloroplasts, primarily uninterrupted photosynthesis. A study of the seasonal dynamics of photosynthetic pigments in pine needles from permafrost ecosystems revealed a decline in chlorophylls *a*, *b*, and carotenoids during the springtime. The authors attribute this to the reactivation of the photosynthetic electron transport chain [44].

We found that the highest absolute content of FAs in the PL fraction was maintained in spring compared to what was observed in winter. Most likely, during the change from winter to spring, as air temperature warms up (April–May), shifts occur in the ratio of biosynthesis to degradation of fatty acids towards their active biosynthesis in cell membranes in *L. cajanderi* shoots at the start of the vegetation period. Such changes in the proportion of FAs may also be caused by the activation of FA biosynthesis in the mechanisms providing *L. cajanderi* shoot tissues with higher resistance to spring frost at the start of needle budding and activating subsequent photosynthetic processes. *L. cajanderi* is known to have the highest rate of summer transpiration of any conifer, exceeding that of evergreens by 2–3 times [45].

Apparently, in this context, *L. cajanderi* sheds needles at the onset of frost, enhancing its cryotolerance, including due to the reduced water loss as a result of winter transpiration. In addition to the physiological mechanisms of adaptation, *L. cajanderi* undergoes metabolic modifications in preparation for dormancy, particularly, changes in the composition and contents of FAs, both in membrane lipids (Table 1) and in total lipids (Figure 7). We found that the further increase in the proportion of UFAs in membrane lipids occurred in autumn and coincided with the predominant outflow of metabolites and the preparation of *L. cajanderi* for dormancy. In this period, there was a drop in the average daily air temperature and a decrease in solar radiation. Simultaneously, an increase in ALA up to 40.1% and LA up to 16.3% was observed. There were significant changes in the functioning of Δ9-desaturase (SDR), whose activity declined significantly in summer compared to spring, but rose again in autumn and winter. In addition, the relatively high DBI index values during autumn and the consistently high ODR and LDR desaturase activity in *L. cajanderi* shoots over the annual cycle may contribute to the phase stability of lipids during the period of low negative temperatures in the cryolithozone. Studies of membrane lipids in *L. cajanderi* shoots revealed that an increase in membrane fluidity due to an increase in FA levels during this period promotes adaptive transformations and the preservation of cellular structures as the conifer enters biological dormancy.

Plant tolerance to cold stress may also depend on the degree of fatty acid saturation in membrane lipids. It is shown in Ref. [46] that the accumulation of unsaturated FAs in cell membranes of the cold-tolerant evergreen shrub *Ephedra monosperma*, especially polyunsaturated ones, improves the plant’s resistance to low-temperature damage. In addition, the high levels of PUFAs in the membranes, which increase in the overwintering ferns of *Asplenium scolopendrium* as the cold season approaches, seem to contribute to the successful overwintering of the plants [47]. These data confirm that cold-tolerant plants can adapt to low temperatures by increasing unsaturated fatty acids to maintain the bilayer membrane structure and ensure membrane fluidity in varying environmental conditions.

Table 1 shows that there is an inverse correlation between the unsaturated FA concentration and air temperature in the spring–winter period, which is reflected in a lower unsaturation index (UI).

In conclusion, the presence of significant diversity in the composition and content of membrane lipids and their FAs, as well as the presence and high levels of ∆5-UPIFA and VLCFA in the shoots of *L. cajanderi* in autumn and winter, indicates that these metabolites significantly influence the general mechanisms involved in the formation of cold tolerance in conifers. It is therefore hypothesised that these molecules may serve as molecular markers of cold tolerance, with potential applications in the breeding of high-yielding and stress-adapted cultivated plants, such as fruit trees, in biotechnology, and in the maintenance of the stability of boreal ecosystems in the forestry practice.

## 4. Materials and Methods

### 4.1. Plant Material and Growth Conditions

The objects of this study were one-year-old shoots (without needles) of Kajander larch (*Larix cajanderi* Mayr) growing in natural conditions in the cryolithozone (forest park zone of the Botanical Garden of IBPC SB RAS, near Yakutsk, Republic of Sakha (Yakutia), Russia 62° N, 129° E). The trees were 45–50 years old. During the experimental period, the climate of the cryolithozone of Yakutia was characterized by extreme conditions: the average monthly air temperature in January was −36.9 °C, in July +19.9 °C; the average precipitation was 237 mm. Temperature values in the region were monitored at http://meteo.ru/data (accessed on 17 September 2024). In general, the climatic parameters during the sampling period did not differ from the long-term average.

### 4.2. Field Experiment

To study the seasonal dynamics of lipids, material was collected at least once a month all year round. We collected 5–7 samples of *L. cajanderi* shoots from three trees. Samples were collected in the morning (9:00–11:00). The samples were immediately fixed in liquid nitrogen and transported in Dewar vessels to the laboratory. The samples were stored in the freezer at −80 °C prior to analysis (Panasonic, Tokyo, Japan).

### 4.3. Lipid Extraction

For lipid extraction, *L. cajanderi* shoots (300 mg) were placed in glass vials that contained 10 mL of Folch solution. Ionol (0.00125 g per 100 mL of mixture) was added to the mixture as an antioxidant. The contents of the vials were thoroughly mixed and allowed to stand undisturbed for 24 h at −20 °C until complete diffusion of the lipids into the solvent. The plant material was then mechanically disintegrated by grinding in a mortar. The resulting solution was quantitatively transferred to a separating funnel via a filter. The mortar and filter were washed three times with the same solvent mixture. In order to separate the non-lipid components, 5 mL of distilled water was added.

### 4.4. PL Composition Analysis by HPTLC–UV/Vis/FLD Method

The lower chloroform fraction was separated for the analysis of PL. Chloroform (high purity grade, stabilized with 0.005% amylene) was removed from the lipid extract under vacuum using the UL-2000 rotary evaporator (Ulab, Shanghai, China). The separation of PL was achieved through the utilization of an HPTLC method on silica gel plates 60 (10 × 10 cm) (Merck, Darmstadt, Germany). The plate was positioned within a chamber containing a mixture of chloroform, methanol, and water (65:25:4 *v*/*v*/*v*). The development of the chromatograms was then conducted in 10% sulfuric acid in methanol, followed by heating at 140 °C. Subsequently, the plate was sprinkled with reagent using a sprayer (Lenchrom, Saint Petersburg, Russia) connected to a JAS 1202 compressor (JAS-AIR, Hong Kong, China). The plate was then dried and transferred to a desiccator, where it was heated for 20 min at 140 °C. The PL content was determined densitometrically using the Sorbfil TLC View (Imid, Krasnodar, Russia) and documented at white light illumination (Vis), UV 254 nm, and FLD 366 nm. The content of individual classes of lipids in chromatograms was calculated using the Sorbfil TLC Videodensitometer 2.3 program. The lipids were identified using standards for target components and specific reagents for individual functional groups [48]. The calculation of the content of individual classes of lipids in chromatograms was carried out using the Sorbfil TLC View 2.3 program using standards PC solutions (Larodan, Solna, Sweden), lysoPC (Sigma, St. Louis, MO, USA), PI (Sigma, St. Louis, MO, USA), PS (Sigma, St. Louis, MO, USA), PG (Sigma, St. Louis, MO, USA), DPG (Sigma, St. Louis, MO, USA), PE (Sigma, St. Louis, MO, USA), PA (Sigma, St. Louis, MO, USA), DGDG (Avanti Polar Lipids, Alabaster, AL, USA), SQDG (Avanti Polar Lipids, Alabaster, AL, USA), MGDG (Plant) (Avanti Polar Lipids, Alabaster, AL, USA), and Rf values from the literary sources [48].

### 4.5. Determination of FA Content in PLs

To study the composition of FAs, the PL fraction was isolated using one-dimensional TLC. The PL spots at the start were scraped out and eluted with a mixture of chloroform: methanol (2:1), then the extract was evaporated and methanolysis was carried out using 1% H_2_SO_4_ in methanol for 30 min at 80 °C. The obtained methyl esters were analyzed by GC–FID on an Agilent 6890N gas chromatograph (Agilent Technologies, Santa Clara, CA, USA) with an HP-INNOWAX, 30 m × 0.25 mm × 0.50 μm capillary column. The temperature program used was from 100 to 240 °C at a rate of 5–6 °C/min. Identification was carried out using the Supelco 37 Component FAME Mix mixture of individual fatty acid methyl esters (Supelco, Bellefonte, PA, USA).

### 4.6. GC–MS Analysis of the Composition and Contents of FAs in Total Lipids

The lipid extraction, lipid transesterification (methylation), and methyl esters purification methods used in this study are described in detail elsewhere [46]. The FAME analysis was performed by GC–MS using the 5973/6890N MSD/DS gas chromatograph–mass spectrometer (Agilent Technologies, Santa Clara, CA, USA). To identify FAs, the NIST 08 mass spectral library and the Christie FAME mass spectral archive were used [49,50]. In some cases, the calculation of the equivalent length of the carbon chain (ECL) was used.

### 4.7. Statistical Processing

The experiments were performed at least in 3–4 independent repetitions (*n* = 3–4). The data obtained are presented as the arithmetic mean (M), and the spread of values was expressed as a standard deviation (±S.D.). Normality of the distribution was tested using the Shapiro–Wilk criterion. To test for differences between the values following one-way ANOVAs, the Newman–Keuls test was used. Differences were considered statistically significant at *p* ≤ 0.05. In the figures and tables, different letters are used to indicate significant differences.

Statistical calculations were performed using the SigmaPlot 12.5 software package.

## 5. Conclusions

Thus, we first studied the lipid profile in the annual cycle of coniferous plants using HPTLC–UV/Vis/FLD, TLC–GC/FID, and GC–MS methods on the example of shoots of *L. cajanderi* growing in the cold climate of Yakutia. The patterns of seasonal changes, significant diversity in the composition and content of membrane lipids and their FAs, together with the presence and high levels of ∆5-UPIFA and VLCFA in *L. cajanderi* shoots in autumn and winter, indicate that these metabolites significantly influence the general mechanisms involved in the development of cold resistance in coniferous plants native to the cryolithozone. The results obtained may provide valuable information for identifying the contribution of membrane lipids and their fatty acids with specific functions to the stability of cell membranes in woody plants during their environmental adaptation to the low temperatures of the cryolithozone. It is hypothesised that, in the context of a warming climate, there is an increasing need to identify plant species and genotypes that are optimally adapted to enhance frost and drought tolerance, as well as resistance to pests and diseases. Consequently, alterations in the lipid composition of tree membranes, including conifers, can be regarded as molecular markers of cold tolerance. This has significant implications for the selection of high-yielding and stress-adapted crops, such as fruit trees, in biotechnology, as well as for maintaining the stability of boreal ecosystems in the forestry practice.

## Figures and Tables

**Figure 1 ijms-26-00164-f001:**
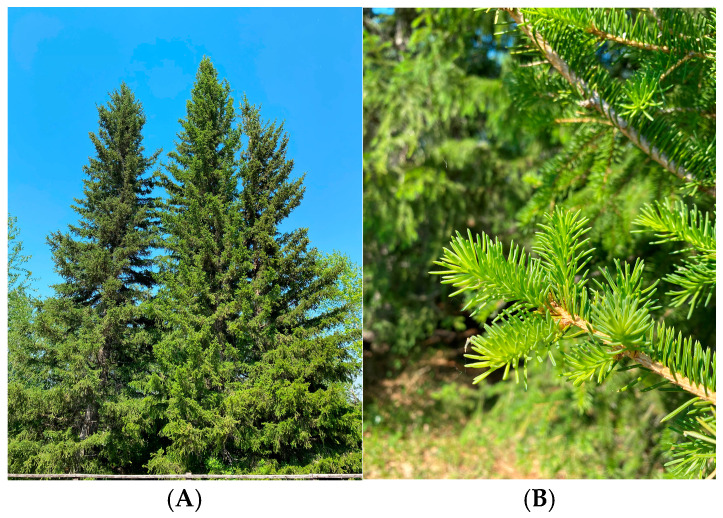
*Larix cajanderi* Mayr (**A**), growing on the territory of the Botanical Garden in the cryolithozone of Yakutia (photo taken on June 2024); *Larix cajanderi* Mayr (**B**), shoots of the current year (photo taken on June 2024).

**Figure 2 ijms-26-00164-f002:**
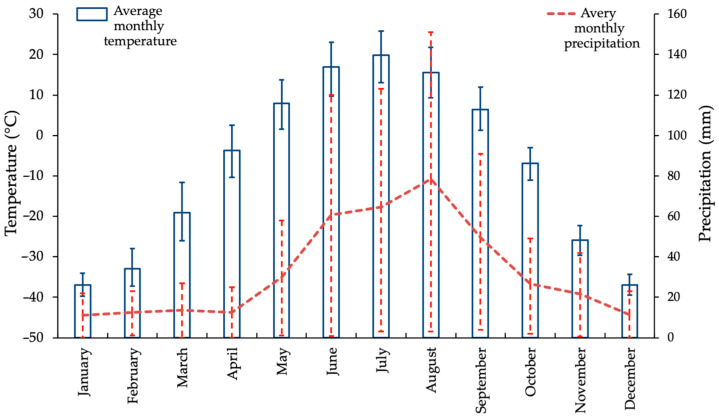
The rate of average monthly air temperature (°C) and average precipitation (mm) (red dashed lines) in the cryolithozone.

**Figure 3 ijms-26-00164-f003:**
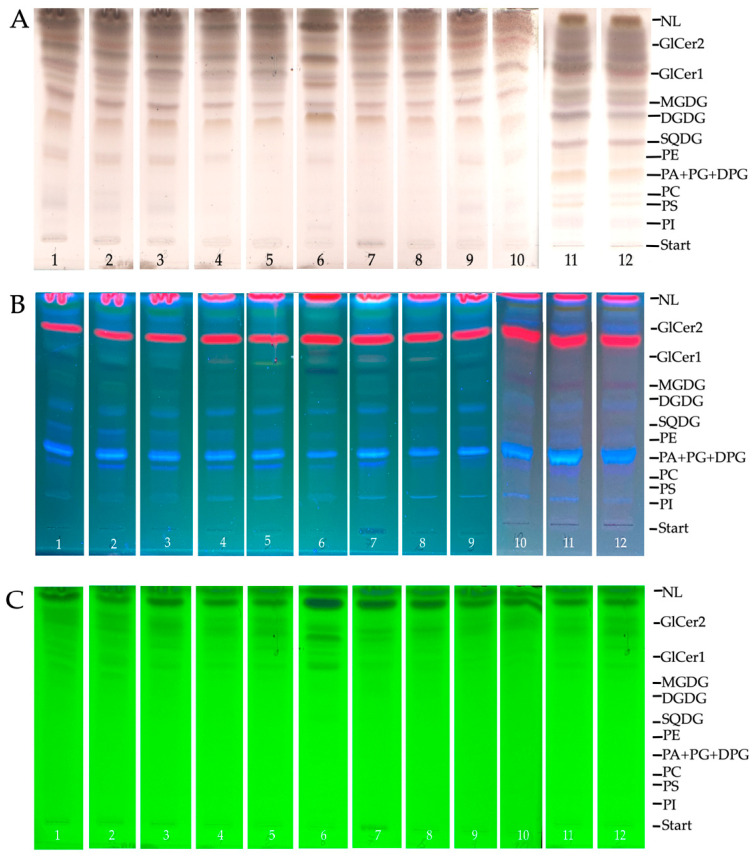
HPTLC profile of polar lipids in *L. cajanderi* shoots in a one-year cycle. Images were taken at (**A**) white light after derivatization; (**B**) 366 nm; (**C**) 254 nm; Track 1—January, Track 2—February, Track 3—March, Track 4—April, Track 5—May, Track 6—June, Track 7—July, Track 8—August, Track 9—September, Track 10—October, Track 11—November, Track 12—December. Different colors represent different lipid classes.

**Figure 4 ijms-26-00164-f004:**
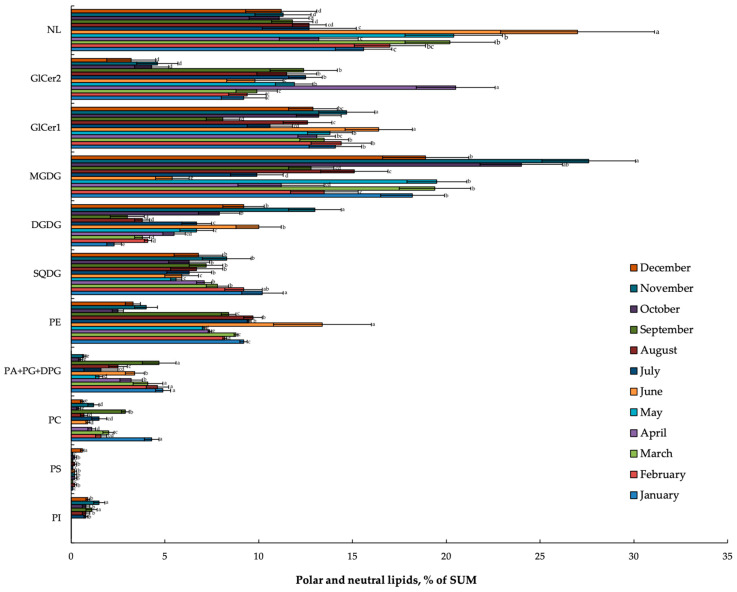
Seasonal changes in the composition of membrane lipids in *L. cajanderi* shoots in the cryolithozone of Yakutia (% of lipid structures). The samples were selected at various points throughout the annual cycle. PC—phosphatidylcholine; PI—phosphatidylinositol; PE—phosphatidylethanolamine; PG—phosphatidylglycerol; PA—phosphatidic acid; DPG—diphosphatidylglycerol; GlCer—glycoceramide, DGDG—digalactosyldiglyceride, MGDG—monogalactosyldiglyceride, SQDG—sulfoquinovosyldiacylglycerol, PS—phosphatidylserine, NL—neutral lipid. Different letters above the bars indicate statistically significant differences at *p*-value < 0.05 (*t*-test).

**Figure 5 ijms-26-00164-f005:**
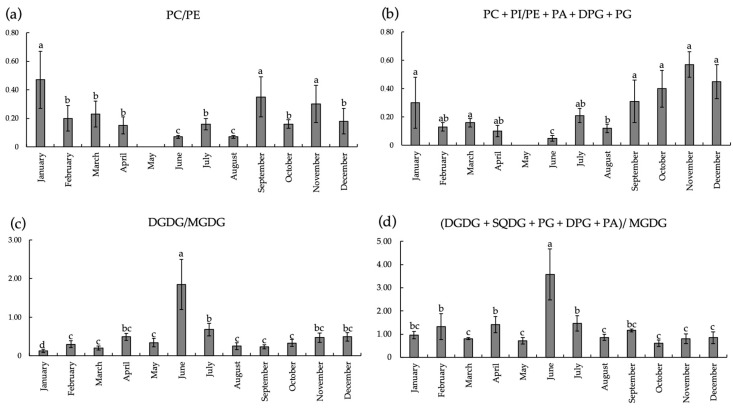
Dynamics of PC/PE, PC + PI/PE + PA + DPG + PG, DGDG/MGDG, (DGDG + SQDG + PG + DPG + PA)/MGDG ratios in *L. cajanderi* shoots (% of lipid structures). Values are presented as the means ± SEMs. The significance of differences between the compared mean values was assessed Kruskal–Wallis ANOVA by ranks (*p* < 0.05). Different superscript letters indicate significant differences of analyzed parameters.

**Figure 6 ijms-26-00164-f006:**
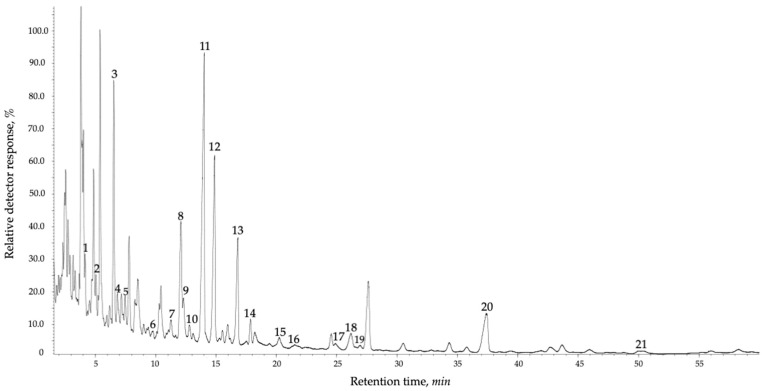
GC–MS chromatogram of fatty acid methyl esters (FAME) in the shoots of *L. cajanderi* growing in the cryolithozone of Yakutia. Peaks: 14:0 (1), 15:0 (2), 16:0 (3), 16:1n−9 (4), 17:0–a (5), 18:1–i (6), 18:0 (7), 18:1(n−9) (8), 18:1n−7 (9), 18:2∆5.9 (taxoleic) (10), 18:2n−6 (LA) (11), 18:3∆5.9.12 (pinolenic) (12), 18:3n−3 (ALA) (13), 18:4(∆5.9.12.15) (coniferonic) (14), 20:0 (15), 20:1n−9 (16), 20:2n−6 (17), 20:3(5.11.14) (sciadonic) (18), 20:3(∆7.11.14) (bis-homo-linolenic) (19), 22:0 (20), 23:0 (21).

**Figure 7 ijms-26-00164-f007:**
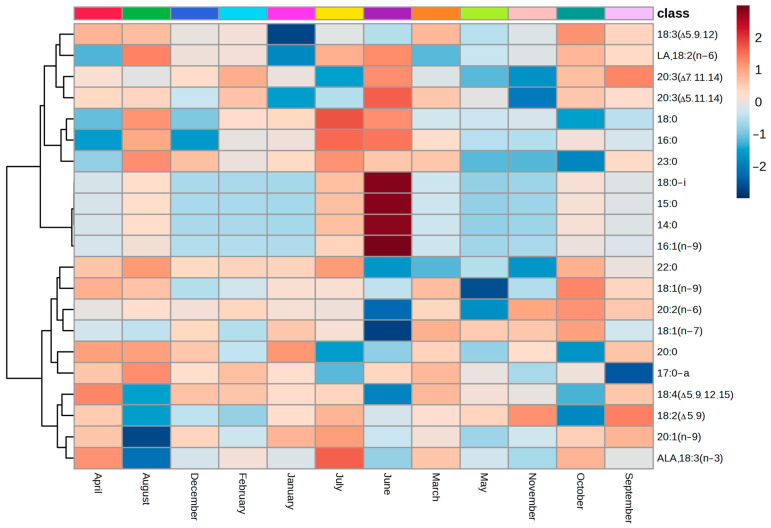
Heat map of the seasonal dynamics of FAs in the composition of total lipids in *L. cajanderi* shoots in the cryolithozone of Yakutia. The samples were collected at various points throughout the annual cycle.

**Figure 8 ijms-26-00164-f008:**
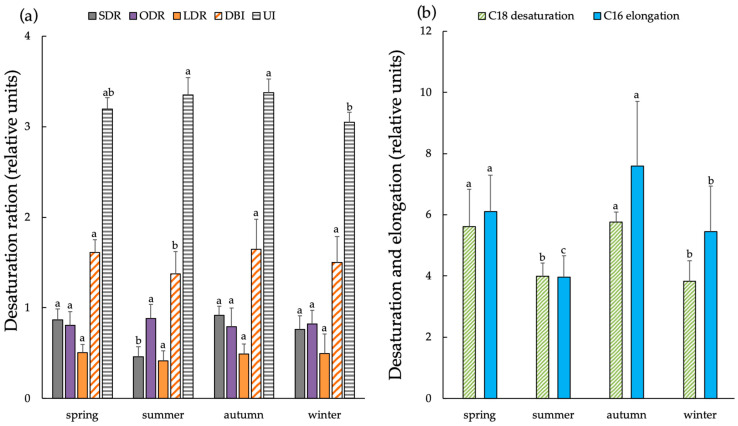
Desaturation ratios (**a**) and desaturation and elongation (**b**) of *L. cajanderi* shoots in spring, summer, autumn, and winter. SDR—stearic desaturation ratio; ODR—oleic desaturation ratio; LDR—linoleic desaturation ratio; DBI—double-bond index; UI—unsaturation index. Values are presented as the means ± SEMs. The significance of differences between the compared mean values was assessed using Kruskal–Wallis ANOVA by ranks (*p* < 0.05). Different superscript letters indicate significant differences of analyzed parameters.

**Figure 9 ijms-26-00164-f009:**
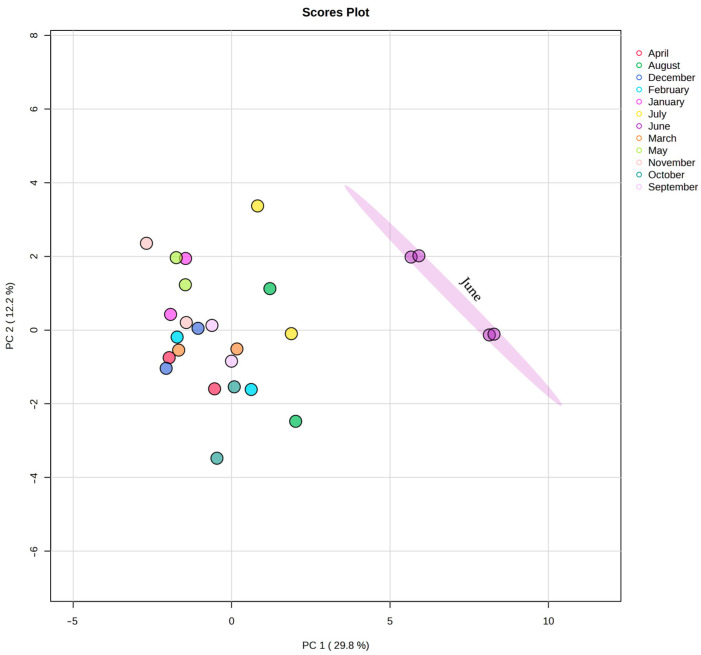
PCA score plot of principal component analysis of FAs profiles in *L. cajanderi* shoots during the annual cycle.

**Table 1 ijms-26-00164-t001:** Seasonal changes in the fatty acids of polar lipids in *L. cajanderi* shoots (% of sum).

FAs	Seasonal			
	Spring	Summer	Autumn	Winter
12:0	0.60 ± 0.09 ^c^	0.97 ± 0.10 ^b^	1.15 ± 0.02 ^a^	0.15 ± 0.03 ^d^
14:0	1.70 ± 0.30 ^a^	1.01 ± 0.10 ^b^	1.11 ± 0.67 ^ab^	1.02 ± 0.13 ^b^
16:0	15.82 ± 1.50 ^b^	18.07 ± 0.80 ^a^	12.87 ± 1.10 ^c^	13.02 ± 2.51 ^c^
18:0	2.78 ± 0.70 ^a^	2.05 ± 1.10 ^a^	1.22 ± 0.47 ^ab^	1.17 ± 0.30 ^ab^
18:1n−7	1.48 ± 0.20 ^d^	8.96 ± 0.80 ^a^	4.35 ± 0.60 ^b^	2.37 ± 0.23 ^c^
18:1n−9	1.97 ± 0.10 ^b^	7.06 ± 0.60 ^a^	1.79 ± 0.29 ^b^	1.55 ± 0.44 ^b^
18:2∆5.9	2.00 ± 0.85 ^b^	3.81 ± 0.92 ^a^	4.45 ± 0.65 ^a^	4.59 ± 0.53 ^a^
18:2n−6	17.18 ± 1.30 ^b^	11.15 ± 1.61 ^c^	16.27 ± 1.49 ^b^	20.07 ± 1.46 ^a^
18:3∆5.9.12	2.45 ± 0.92 ^a^	1.81 ± 0.98 ^a^	1.71 ± 0.74 ^a^	1.62 ± 0.82 ^a^
18:3n−3	38.92 ± 2.12 ^a^	32.40 ± 1.67 ^b^	40.78 ± 2.63 ^a^	39.92 ± 1.59 ^a^
20:0	1.24 ± 0.55 ^a^	0.27 ± 0.04 ^d^	0.50 ± 0.11 ^c^	0.79 ± 0.14 ^b^
20:3∆5.11.14	2.86 ± 1.12 ^a^	2.62 ± 0.90 ^a^	3.03 ± 0.65 ^a^	2.90 ± 0.77 ^a^
22:0	6.73 ± 1.10 ^a^	4.73 ± 1.32 ^b^	6.01 ± 1.12 ^a^	6.09 ± 1.11 ^a^
23:0	4.27 ± 0.72 ^a^	5.08 ± 0.50 ^a^	4.74 ± 0.95 ^a^	4.74 ± 0.91 ^a^
∆5-UPIFA *	7.30 ± 2.10 ^b^	8.24 ± 2.82 ^ab^	9.19 ± 2.01 ^a^	9.11 ± 1.92 ^a^
SFA **	33.14 ± 2.40 ^a^	32.19 ± 2.20 ^a^	27.61 ± 1.60 ^b^	26.98 ± 3.60 ^b^
UFA ***	66.85 ± 5.60 ^b^	67.80 ± 4.40 ^b^	72.38 ± 4.80 ^a^	73.02 ± 6.10 ^a^
Sum FAs (μg/g)	27.32 ± 5.20 ^a^	17.34 ± 2.30 ^b^	20.05 ± 2.70 ^b^	14.46 ± 1.67 ^c^

* ∆5-UPIFA—delta5-unsaturated polymethylene-interrupted fatty acids. ** SFA—saturated FAs. *** UFA—unsaturated FAs. Values in the rows marked with the same letters were not significantly different at *p* < 0.05.

## Data Availability

The data used to support the findings of this study can be made available by the corresponding author upon request.

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
