# Peer review of "Lipid Profile of Larix cajanderi Mayr in Adaptation to Natural Conditions in the Cryolithozone"

_ijms, 2024, doi:10.3390/ijms26010164_

Round 1
Reviewer 1 Report
Comments and Suggestions for Authors
Review on „Lipid Profile of Larix Cajanderi Mayr in Adaptation to Extreme Cold in the Cryolithozone”
Plants have evolved a variety of adaptations to survive and thrive in cold environments. These adaptations help them cope with freezing temperatures, limited water availability, and reduced sunlight during the colder months. Adaptations like physical, behavioral, biochemical, and structural adaptations, morphological adjustments, enable plants to survive in diverse cold environments, from temperate forests to polar tundra, ensuring their resilience and reproduction even in challenging climates. The authors examined the effect of cold temperature on the lipid profile of Larix Cajanderi Mayr.
I do not quite see why it is interesting to study but I have not studied pines so it may be relevant for a narrow community. This study looks like rather a theoretical study than a practical, useable study. I suggest adding more of a practical emphasis to this study.
I suggest adding a more reliable conclusion or future outlook of the study to the Conclusions section, e.g. Because of the unpredictable weather conditions, the research data of this study can be used in the further examination of fruit trees’ cold adaptation as well. Although global warming leads to rising average temperatures, some polar areas paradoxically experience colder weather due to disrupted atmospheric patterns. Nevertheless, this polar weather effect can be appear more frequently in cities as well. For example in 2021, Texas experienced an unusual cold snap linked to polar vortex shifts. While tropical climates face intensifying heat and its cascading effects, colder weather events in the poles and temperate regions highlight the complex dynamics of climate change. A global approach to mitigation and adaptation is essential to address these varied challenges.
Specific comments:
Title:
I suggest changing the title. Extreme cold means lower than the average low temperature. If the average winter temperature is -30 °C degree somewhere, it is not an extreme cold weather at that region. Extreme cold compared to the warmer climate.
Abstract:
Please add the clear goal and hypothesis of the study. „This study is the first to have investigated…” can’t be a motivation of a scientific investigation. Maybe adding the hypothesis of the study would be easier…and in the Discussion and Conclusion sections the authors can confirm or deny their hypothesis.
Keywords:
Please arrange the keywords in alphabetical order.
Introduction:
Line 42: Please add the full name of every abbreviation when they are firstly presented in the manuscript despite them being presented in the Abstract.
Lines 43-46. In my opinion, the examination of cold adaptation has more importance in case of crops and other edible plants. Why is it so important to examine the cold adaptation of woody plants like pines?
Lines 51-61. Yes, these are true for crops (Dudareva et al., 2024 – white) or herbaceous plants. What about the woody plants?
Lines 94, 96: “low-temperature resistance in conifers” so did the author examine the low-temperature resistance or extreme cold? The title contains extreme cold. The meaning of “extreme cold “ is different in tropical and polar areas. For example +15°C degree can be an extreme cold in the Amazon, while -90°C degree (e,g, Antarctica at -93.3°C) is an extreme cold in polar regions. Please check and use extreme cold or low-temperature consistently in the manuscript.
Lines 100-103. Please rewrite. See my previous comment about the goal and hypothesis of the study above.
Author Response
Response to Reviewer 1
Authors: We would like to thank the esteemed Reviewers for the interest in our work and careful analysis of the results.
Reviewer: Review on „Lipid Profile of Larix Cajanderi Mayr in Adaptation to Extreme Cold in the Cryolithozone”
Plants have evolved a variety of adaptations to survive and thrive in cold environments. These adaptations help them cope with freezing temperatures, limited water availability, and reduced sunlight during the colder months. Adaptations like physical, behavioral, biochemical, and structural adaptations, morphological adjustments, enable plants to survive in diverse cold environments, from temperate forests to polar tundra, ensuring their resilience and reproduction even in challenging climates. The authors examined the effect of cold temperature on the lipid profile of Larix Cajanderi Mayr.
I do not quite see why it is interesting to study but I have not studied pines so it may be relevant for a narrow community. This study looks like rather a theoretical study than a practical, useable study. I suggest adding more of a practical emphasis to this study.
I suggest adding a more reliable conclusion or future outlook of the study to the Conclusions section, e.g. Because of the unpredictable weather conditions, the research data of this study can be used in the further examination of fruit trees’ cold adaptation as well. Although global warming leads to rising average temperatures, some polar areas paradoxically experience colder weather due to disrupted atmospheric patterns. Nevertheless, this polar weather effect can be appear more frequently in cities as well. For example in 2021, Texas experienced an unusual cold snap linked to polar vortex shifts. While tropical climates face intensifying heat and its cascading effects, colder weather events in the poles and temperate regions highlight the complex dynamics of climate change. A global approach to mitigation and adaptation is essential to address these varied challenges.
Authors: We would like to thank the reviewer for their valuable comments that we have used to improve the quality of our manuscript. We appreciate your input and agree with all your comments. We have tried to make the necessary corrections to address the shortcomings and improve the overall quality of the paper. Corrections made in the text are highlighted in green. Below are our detailed responses to each comment. We have added the hypothesis to the manuscript, reworded the objective, and modified the discussion and conclusion based on your comments.
Specific comments:
Reviewer: Title:
I suggest changing the title. Extreme cold means lower than the average low temperature. If the average winter temperature is -30 °C degree somewhere, it is not an extreme cold weather at that region. Extreme cold compared to the warmer climate.
Authors: Thank you for your advice. We have changed the title of the manuscript. Lipid Profile of Larix Cajanderi Mayr in Adaptation to Natural Condition in the Cryolithozone
Reviewer: Abstract:
Please add the clear goal and hypothesis of the study. „This study is the first to have investigated…” can’t be a motivation of a scientific investigation. Maybe adding the hypothesis of the study would be easier…and in the Discussion and Conclusion sections the authors can confirm or deny their hypothesis.
Authors: Thank you for your advice. We have made the changes you suggested to the manuscript.
Reviewer: Keywords:
Please arrange the keywords in alphabetical order.
Authors: Done.
Reviewer:
Introduction:
Line 42: Please add the full name of every abbreviation when they are firstly presented in the manuscript despite them being presented in the Abstract.
Authors: Done.
Reviewer: Lines 43-46. In my opinion, the examination of cold adaptation has more importance in case of crops and other edible plants. Why is it so important to examine the cold adaptation of woody plants like pines?
Authors: The Cajander larch is a coniferous tree with unrivaled frost resistance. Its habitats extend far beyond the Arctic Circle; this species defines the northern boundary of woody plant growth. On the one hand, it is difficult to overestimate the economic value of trees growing in the harsh conditions of the Far North and providing the population of these places with wood for construction, heating, etc. On the other hand, the analysis of the biochemical mechanisms of cold resistance, in particular, the study of lipid adaptation that allows the Cajander larch to survive in these conditions, are of both fundamental and practical interest. Analysis of the composition of lipids and fatty acids in the tissues of this species in the annual cycle will reveal the features of lipid metabolism that enable trees to adapt to the (extremely) low temperatures of the permafrost zone of Yakutia. From a practical point of view, the information obtained can be used to study the mechanisms of adaptation of woody plants, including fruit trees, to cold. It is possible to use this information in the selection of molecular markers of cold resistance of woody plants.
Reviewer: Lines 51-61. Yes, these are true for crops (Dudareva et al., 2024 – white) or herbaceous plants. What about the woody plants?
Authors: Done. Compared with woody plants. Thanks for the advice.
Reviewer: Lines 94, 96: “low-temperature resistance in conifers” so did the author examine the low-temperature resistance or extreme cold? The title contains extreme cold. The meaning of “extreme cold “ is different in tropical and polar areas. For example +15°C degree can be an extreme cold in the Amazon, while -90°C degree (e,g, Antarctica at -93.3°C) is an extreme cold in polar regions. Please check and use extreme cold or low-temperature consistently in the manuscript.
Authors: low-temperature resistance in conifers
Reviewer: Lines 100-103. Please rewrite. See my previous comment about the goal and hypothesis of the study above.
Authors: Done.
Reviewer 2 Report
Comments and Suggestions for Authors
Comment
The paper presents Lipid Profile of Larix Cajanderi Mayr in Adaptation to Extreme Cold in the Cryolithozone. It discusses investigated seasonal changes in the 16 lipid profile of L. cajanderi within the annual cycle using HPTLC-UV/Vis/FLD, TLC-GC/FID and 17 GC-MS techniques. I believe the authors addressed an important topic but it still needs further details and comprehension, which can be provided with incorporating minor revisions.
1. In methodology, I found plagiarism. Please paraphrase it.
2. In line 110, “One-year-old shoots without needles (Figure 1B) were sampled once a month 110 during the year.”, how to know that the shoots is one year old?
3. Line 147-148 is too long. Authors can separate it into 2 paragraphs
4. Line 343-354, authors need to compare their results with previous studies related to this matter. Of it is similar or maybe not similar. Please give a comprehensive elucidation.
5. Similar with previous comments, line 354-372, line 373-379, , please compare your results with previous studies.
6. Line 388-395, please add citations
7. Line 440-442 can be put together with previous paragraph. Then, authors need to make this sentence relate to previous paragraph.
8. Authors need to add more references with the updated papers ( the last 5 years)
9. Keywords should be alphabetically orders
10. Line 32-37 can be put together with the second paragraph. Please check again the introduction. Small paragraph can be put together with other paragraph
Author Response
Response to Reviewer 2
Authors: We would like to thank the reviewer for their valuable comments that we have used to improve the quality of our manuscript. We appreciate your input and agree with all your comments. We have tried to make the necessary corrections to address the shortcomings and improve the overall quality of the paper. Corrections made in the text are highlighted in green. Below are our detailed responses to each comment. We have added the hypothesis to the manuscript, reworded the objective, and modified the discussion and conclusion based on your comments.
Reviewer: Comment
The paper presents Lipid Profile of Larix Cajanderi Mayr in Adaptation to Extreme Cold in the Cryolithozone. It discusses investigated seasonal changes in the 16 lipid profile of L. cajanderi within the annual cycle using HPTLC-UV/Vis/FLD, TLC-GC/FID and 17 GC-MS techniques. I believe the authors addressed an important topic but it still needs further details and comprehension, which can be provided with incorporating minor revisions.
Reviewer: 1. In methodology, I found plagiarism. Please paraphrase it.
Authors: Done. Thanks for the advice.
Reviewer: 2. In line 110, “One-year-old shoots without needles (Figure 1B) were sampled once a month 110 during the year.”, how to know that the shoots is one year old?
Authors: Larch Cajanderi is a coniferous plant whose needles fall at the end of the season. As a result, in the spring (late April-May) at the beginning of active growth, when the average daily air temperature goes over 100C, trees begin to form new annual shoots, on which needles appear and grow during the vegetation cycle. Visually, one-year shoots of larch differ from other shoots by their light yellow color and the absence of a hard cortex. After the needles fall in September, the shoots of the current year are preserved until the beginning of the next season. The one-year shoot of larch considered in this work as the object of study is a shoot of the first year on which buds have not yet formed, collected for analysis monthly throughout the year. As a result, in spring (late April-May) at the beginning of active growth, when the average daily air temperature goes over 100C, trees begin to form new annual shoots on which needles appear and grow during the vegetation cycle. Visually, one-year shoots of larch differ from other shoots by their light yellow color and the absence of a hard cortex. After the needles fall in September, the shoots of the current year are preserved until the beginning of the next season. The one-year shoot of larch considered in this work as the object of study is a shoot of the first year on which buds have not yet formed, collected for analysis monthly throughout the year.
Reviewer: 3. Line 147-148 is too long. Authors can separate it into 2 paragraphs
Authors: Done.
Reviewer: 4. Line 343-354, authors need to compare their results with previous studies related to this matter. Of it is similar or maybe not similar. Please give a comprehensive elucidation.
Authors: Done. Thank you for your valuable advice. We compared our results with previous studies and added them to the manuscript.
Reviewer: 5. Similar with previous comments, line 354-372, line 373-379, , please compare your results with previous studies.
Authors: Done. Thank you for your valuable advice. We compared our results with previous studies and added them to the manuscript.
Reviewer: 6. Line 388-395, please add citations
Authors: Done.
Reviewer: 7. Line 440-442 can be put together with previous paragraph. Then, authors need to make this sentence relate to previous paragraph.
Authors: Done.
Reviewer: 8. Authors need to add more references with the updated papers ( the last 5 years)
Authors: Done. We reviewed and added the latest research in this area.
Reviewer: 9. Keywords should be alphabetically orders
Authors: Done
Reviewer: 10. Line 32-37 can be put together with the second paragraph. Please check again the introduction. Small paragraph can be put together with other paragraph
Authors: Done
Authors: We would like to thank the esteemed Reviewers for the interest in our work and careful analysis of the results.
Round 2
Reviewer 1 Report
Comments and Suggestions for Authors
Thank you for the corrections.
Happy Holidays!